# Gustatory and Saliva Secretory Dysfunctions in COVID-19 Patients with Zinc Deficiency

**DOI:** 10.3390/life12030353

**Published:** 2022-02-28

**Authors:** Hironori Tsuchiya

**Affiliations:** School of Dentistry, Asahi University, Mizuho, Gifu 501-0296, Japan; tsuchi-hiroki16@dent.asahi-u.ac.jp; Tel.: +81-58-329-1263

**Keywords:** COVID-19, gustatory dysfunction, saliva secretory dysfunction, pathogenic speculation, zinc deficiency, carbonic anhydrase, metallothionein, zinc transporter

## Abstract

Given the ever-progressing studies on coronavirus disease 2019 (COVID-19), it is critical to update our knowledge about COVID-19 symptomatology and pathophysiology. In the present narrative review, oral symptoms were overviewed using the latest data and their pathogenesis was hypothetically speculated. PubMed, LitCovid, ProQuest, and Google Scholar were searched for relevant studies from 1 April 2021 with a cutoff date of 31 January 2022. The literature search indicated that gustatory dysfunction and saliva secretory dysfunction are prevalent in COVID-19 patients and both dysfunctions persist after recovery from the disease, suggesting the pathogenic mechanism common to these cooccurring symptoms. COVID-19 patients are characterized by hypozincemia, in which zinc is possibly redistributed from blood to the liver at the expense of zinc in other tissues. If COVID-19 induces intracellular zinc deficiency, the activity of zinc-metalloenzyme carbonic anhydrase localized in taste buds and salivary glands may be influenced to adversely affect gustatory and saliva secretory functions. Zinc-binding metallothioneins and zinc transporters, which cooperatively control cellular zinc homeostasis, are expressed in oral tissues participating in taste and saliva secretion. Their expression dysregulation associated with COVID-19-induced zinc deficiency may have some effect on oral functions. Zinc supplementation is expected to improve oral symptoms in COVID-19 patients.

## 1. Introduction

Patients infected with severe acute respiratory syndrome coronavirus 2 (SARS-CoV-2) are well known to present with fever, cough, dyspnea, myalgia, fatigue, myocarditis, cardiomyopathy, arrhythmia, cardiac arrest, anorexia, nausea, and diarrhea [1,2]. In addition to these clinical manifestations, chemosensory disorders such as anosmia and ageusia have been frequently reported by patients with coronavirus disease 2019 (COVID-19) caused by SARS-CoV-2. However, COVID-19 ageusia is not necessarily accompanied by nasal obstruction and rhinitis associated with smell loss [3,4]. Ageusia is more prevalent than anosmia in some of the studies on COVID-19 symptomatology [3,5].

It is also becoming evident that the symptoms specific to oral tissues and functions are closely associated with COVID-19 [6,7,8]. One of them is gustatory dysfunction [8,9,10], consisting of ageusia (complete taste loss) and dysgeusia (taste impairment) that is further classified into mild hypogeusia (or amblygeustia), moderate hypogeusia, and severe hypogeusia. COVID-19 patients may also develop other oral symptoms such as saliva secretory dysfunction, which results in dry mouth, xerostomia (subjective complaint of oral dryness), and hyposalivation (objective reduction of salivary flow rates) [7,10,11].

Oral symptoms in COVID-19 patients have been interpreted by relating to the expression of the receptor for viral cellular entry, disturbance of the renin-angiotensin system, inflammation of the relevant oral tissues, cranial neuropathy, intercurrent diseases, ongoing treatments with certain drugs, etc. [8,11,12,13,14]. However, the pathogenic mechanism underlying them remains to be fully elucidated.

Considering that new information on COVID-19 continues to accumulate, it is critical to update our knowledge about COVID-19 symptomatology and pathophysiology. In the present narrative review, gustatory dysfunction and saliva secretory dysfunction in COVID-19 patients and survivors after recovery from the disease were overviewed using the latest data, followed by characterization of COVID-19 patients with hypozincemia. Subsequently, the pathogenesis of both oral symptoms was hypothetically speculated from the perspective of zinc deficiency that could be induced by SARS-CoV-2 infection. The consequence of zinc deficiency was discussed in association with zinc-metalloenzymes, the activity of which depends on cellular zinc concentrations, and with metallothioneins and zinc transporters, both of which control cellular zinc homeostasis. These zinc-binding proteins are expressed and localized in oral tissues responsible for taste perception and saliva secretion. A better understanding of COVID-19 oral symptoms and their pathogenic mechanism may facilitate the development of the potential therapy for improving them.

## 2. Materials and Methods

PubMed, LitCovid, ProQuest, and Google Scholar were searched for relevant studies from 1 April 2021 with a cutoff date of 31 January 2022. Given the rapid worldwide spread of SARS-CoV-2 infection and the ever-progressing studies on COVID-19, the preprint database medRxiv was also used to retrieve the most up-to-date information. The literature search was conducted by using the following terms and combinations thereof: “COVID-19”, “ageusia”, “dysgeusia”, and “hypogeusia” for overviewing gustatory dysfunction in COVID-19; “COVID-19”, “xerostomia”, “dry mouth”, and “hyposalivation” for overviewing saliva secretory dysfunction in COVID-19; “COVID-19”, “serum zinc”, “hypozincemia”, and “zinc deficiency” for characterizing COVID-19 patients; “COVID-19 pathogenesis”, “zinc status”, “zinc-metalloprotein”, “carbonic anhydrase”, “metallothionein”, and “zinc transporter” for speculating the pathogenesis of COVID-19 oral symptoms. The exclusion criteria were papers that were not published in English and studies that lacked demographic data and did not specify COVID-19 diagnostic methods such as the reverse transcription-polymerase chain reaction (RT-PCR) test. Cited papers in the retrieved articles were further searched for additional references. Collected articles were reviewed by title, abstract, and text for relevance.

## 3. Results

Gustatory dysfunction and saliva secretory dysfunction in the early phase of COVID-19 and after recovery from COVID-19 are firstly overviewed, and then COVID-19 patients are characterized by hypozincemia.

### 3.1. Gustatory Dysfunction

For simplicity, ageusia, dysgeusia, and hypogeusia are collectively expressed as “gustatory dysfunction” in the present study. Results of the literature search included 62 studies [15,16,17,18,19,20,21,22,23,24,25,26,27,28,29,30,31,32,33,34,35,36,37,38,39,40,41,42,43,44,45,46,47,48,49,50,51,52,53,54,55,56,57,58,59,60,61,62,63,64,65,66,67,68,69,70,71,72,73,74,75] about early gustatory dysfunction with a total of 35,870 COVID-19 patients and 38 studies [47,59,60,63,74,76,77,78,79,80,81,82,83,84,85,86,87,88,89,90,91,92,93,94,95,96,97,98,99,100,101,102,103,104,105,106,107,108] about the persistent gustatory dysfunction after recovery from the disease with a total of 14,348 COVID-19 survivors. A meta-analysis was not performed due to heterogeneity in the designs and data of included studies. Studies retrieved for the symptom overview may be potentially biased. In order to evaluate study quality, the risk scores were previously determined by meta-analyses and systematic reviews of Hajikhani et al. [109], Kim et al. [110], Boscutti et al. [111], and Wu et al. [112]. By referring to their results, the overall risks of bias of studies used in the present study appear to be low or moderate. The prevalence of gustatory dysfunction was variable across studies, with wide ranges of 1.0–95.9% in the early phase of COVID-19 and of 1.6–45.0% after at least 28 days from symptom onset or disease diagnosis.

Figure 1A shows the prevalence of early gustatory dysfunction in different COVID-19 cohorts. The prevalence depended on geographical differences between Europe, America, East Asia, and the Middle East (Figure 1B). Pooled prevalence (mean ± SD) was 61.8 ± 12.9% in Europe (Italy, France, Spain, Germany, Switzerland, the UK, Denmark, Sweden, and Poland), 59.1 ± 17.3% in America (the USA and Canada), 14.4 ± 15.7% in East Asia (China, Japan, South Korea, and Singapore) and 39.8 ± 10.7% in the Middle East (Iran, Israel, Turkey, and Qatar). Early gustatory dysfunction in COVID-19 patients is likely to be more prevalent in East Asian cohorts than in European cohorts (*p* < 0.0001), American cohorts (*p* < 0.0001), and Middle Eastern cohorts (*p* < 0.01), whereas there were no significant differences between European, American, and Middle Eastern cohorts. Cirillo et al. [113] indicated that COVID-19 patients exhibit a geographically different pattern of gustatory dysfunction. A comparative study by Kumar et al. [114] also revealed that COVID-19 patients presenting with ageusia show significantly lower percentages in Asia than in Europe and the USA. In a systematic review and meta-analysis of von Bartheld et al. [115], COVID-19 gustatory dysfunction was found to occur sixfold more frequently in Caucasians than in East Asians.

COVID-19 gustatory dysfunction has been suggested to occur depending on gender [28,35,48,51,70]. Its prevalence was 52.6–63.6% and 25.0–39.6% for female and male patients, respectively, in different cohorts [15,35,48,116]. Mercante et al. [55] statistically analyzed the prevalence of chemosensory disorders of COVID-19 patients and demonstrated that severe taste impairment is more prevalent in females than males (female vs. male: odds ratios (OR), 3.16; 95% confidence interval (CI), 1.76–5.67 vs. OR, 2.58; 95% CI, 1.43–4.65). Amorim dos Santos et al. [10] assessed overall oral symptoms of 10,228 patients (including 5770 females and 4288 males) in 19 countries. Their results indicated that gustatory dysfunction is the most common symptom to occur in 45% of the patients (reporting dysgeusia, hypogeusia, and ageusia in decreasing order of prevalence), which was significantly associated with female patients (OR, 1.64; 95% CI, 1.23–2.17). COVID-19 oral symptomatology has suggested the possibility that the prevalence of gustatory dysfunction may vary by other factors such as disease severity and patient age [8,9]. Al-Rawi et al. [117] comparatively determined the prevalence of gustatory dysfunction in different groups of COVID-19 patients. The magnitude of taste alteration increased steeply from the asymptomatic group to the paucisymptomatic group and to the symptomatic group. A systematic review and meta-analysis of chemosensory disorders conducted by Mutiawati et al. [118] indicated a relation between dysgeusia and the severity of COVID-19.

Figure 2 shows the prevalence of persistent gustatory dysfunction in COVID-19 survivors who were followed up for 28–365 days after symptom onset, hospital admission, or disease diagnosis. Gustatory dysfunction can persist for more than a few months with prevalence ranging from 1.6% to 45%, irrespective of geographical or ethnic differences. Gustatory dysfunction persisted in 13.6 ± 11.9% (mean ± SD) of COVID-19 survivors at 29.7 ± 2.1 days follow-up (in 13 studies), 13.8 ± 7.9% at 60.4 ± 2.1 days follow-up (in 9 studies) and 12.7 ± 7.4% at 181.9 ± 3.2 days follow-up (in 7 studies). Moraschini et al. [119] analyzed 8 observational studies to verify the long-term effects of COVID-19 and demonstrated that 14.1% of the subjects have ageusia at mean 67 days follow-up. Even 365 days after symptom onset, gustatory and/or olfactory dysfunction was reported by 12.7–22.0% of COVID-19 survivors in Italian cohorts [107,108]. Ageusia, hypogeusia, and/or dysgeusia may possibly persist for one year after recovery from the disease.

When followed up for 4–6 months after symptom/disease onset or hospital admission, gustatory dysfunction was reported by 1.6–7.3% of COVID-19 survivors in East Asia (China and Japan) [94,101], whereas reported by 5.0–27.1% of COVID-19 survivors in Europe (Italy, Norway, France, Germany, and the UK) [95,97,98,99,100,102,103,104] and the USA [96]. Persistent gustatory dysfunction is less prevalent in Asian cohorts compared with European and American cohorts. Andrew et al. [120] followed up with 114 COVID-19 patients consisting of white (81.6%), Asian (15.8%), black/African/Caribbean (1.3%), and mixed/multiple ethnic (1.3%) after a median of 52 days from symptom onset. Being white positively influenced the recovery time of normal taste.

In an Italian COVID-19 cohort, females needed a longer time to recover normal taste than males [47,81]. When German COVID-19 patients were assessed after a median of 6.8 months from symptom onset, ageusia was reported by 7.9% of females but 3.1% of males, suggesting that the female gender is a higher risk of long-lasting abnormal taste (OR = 0.49; 95% CI, 0.31–0.77) [103]. In a follow-up study of French patients, 17.6% of female and 6.4% of male COVID-19 survivors presented with gustatory/olfactory dysfunction after 7 months from symptom onset [105]. Female patients are likely to cause gustatory sequelae more frequently. The prevalence of persistent COVID-19 gustatory dysfunction may depend on patient age and disease severity, although the significance of these factors is not necessarily clear.

### 3.2. Saliva Secretory Dysfunction

For simplicity, dry mouth, xerostomia, and hyposalivation are collectively expressed as “saliva secretory dysfunction” in the present study. Results of the literature search included 12 studies [15,42,73,93,106,121,122,123,124,125,126,127] about the early saliva secretory dysfunction with a total of 582 COVID-19 patients and about the persistent saliva secretory dysfunction after recovery from the disease with a total of 1049 COVID-19 survivors. Since the number of surveyed subjects was relatively small, all the retrieved studies were used for the symptom overview without a meta-analysis, therefore a part of them may potentially be biased. Aragoneses et al. [7] analyzed the study quality on COVID-19 oral symptoms including saliva secretory dysfunction and found that the risk of bias is low or moderate.

Figure 3A shows the prevalence of saliva secretory dysfunction in the early phase of COVID-19 and persistent saliva secretory dysfunction after recovery from COVID-19 in different cohorts. For assessing the symptom persistence, COVID-19 patients were followed up for 15–253 days after RT-PCR test negativity or hospital discharge. Prevalence (mean ± SD) was 46.5 ± 12.2% and 28.9 ± 21.0% for early and persistent saliva secretory dysfunction, respectively (Figure 3B). The prevalence of xerostomia was different during the active phase of COVID-19 and after the negative results of an RT-PCR test (*p* = 0.002, OR, 23.05; 95% CI, 2.9–182), as reported by Freni et al. [121].

Saliva secretory dysfunction may vary depending on ethnicity, gender, age, and medications. However, the number of the relevant studies is limited as an Asian cohort was assessed by only one study by Chen et al. [15]. Therefore, it is inconclusive whether geographical or ethnic differences influence the prevalence and persistence of saliva secretory dysfunction associated with COVID-19. In the early phase of COVID-19, dry mouth was reported by 34.4% of females and 21.9% of males in 128 Israeli patients [42]. Omezli and Torul [124] objectively evaluated saliva secretory dysfunction of 107 Turkish COVID-19 patients at least 14 days after the completed treatments by measuring the amount of stimulated saliva. Hyposalivation was observed in 18.5% of patients with mild to moderate COVID-19, but saliva flow showed no significant differences between males and females as well as the comparison of an Egyptian cohort [123]. Xerostomia was reported by 14.4% of Israeli COVID-19 survivors at 6.3–8.4 months follow-ups after RT-PCR test negativity, although its prevalence was not statistically different between males and females [106]. When 100 Colombian COVID-19 patients were followed up for 7 months after symptom onset, xerostomia persisted in 25.7% of ambulatory patients with mild COVID-19 and in 37.5% of patients with critical COVID-19, suggesting the association of saliva secretory dysfunction with disease severity [127].

Dry mouth is common in patients with certain comorbidities and medications like antihypertensives [122]. Diabetes mellitus and chronic bronchopulmonary disease are related to reduced saliva secretion [93]. Although some of the COVID-19 patients may have comorbidities and medications, it is certain that saliva secretory dysfunction occurs in a significant number of patients infected with SARS-CoV-2 in the early phase of COVID-19 and continues for at least 8 months after recovery from COVID-19.

### 3.3. Cooccurrence of Oral Symptoms

Out of 12 retrieved studies, 10 studies reported that COVID-19 patients and survivors complain of gustatory dysfunction together with saliva secretory dysfunction. Figure 4A shows the prevalence of both dysfunctions in the early phase of COVID-19 and at 15–253 days follow-ups after hospital discharge, RT-PCR test negativity, or symptom onset. Irrespective of geographical differences, early and persistent gustatory dysfunction can occur simultaneously with xerostomia, dry mouth, and/or hyposalivation. In an Israeli cohort consisting of 128 COVID-19 patients, the prevalence of gustatory dysfunction was correlated with that of dry mouth (*p* = 0.009) [42]. Egyptian COVID-19 patients also presented with both gustatory dysfunction and dry mouth [126]. Ageusia and xerostomia were reported by 53.0% and 26.0% of Colombian COVID-19 survivors, respectively, after a median of 219 days from symptom onset [127].

A moderate correlation was found in prevalence between gustatory dysfunction and saliva secretory dysfunction (Figure 4B). Both dysfunctions are considered to cooccur in the early phase of COVID-19 and consistently persist after recovery from COVID-19, providing an insight into the pathogenic mechanism common to these cooccurring oral symptoms.

### 3.4. Hypozincemia Characterizing COVID-19 Patients

Angiotensin-conversing enzyme 2 (ACE2) receptor, which is primarily responsible for the entry of SARS-CoV-2 into host cells, is abundantly present not only in human taste buds with taste receptors and fungiform papillae taste cells but also in the ducts and acini of human submandibular and parotid glands and minor salivary glands. Therefore, the cytopathic effect of SARS-CoV-2 would damage ACE2-expressing cells during the viral cellular entering process, adversely affecting gustatory and saliva secretory functions. The viral interaction with ACE2 has been cited for the pathogenesis of COVID-19 oral symptoms [7,11,13,14]. Because damaged taste bud cells require weeks to proliferate and recover their functions and the turnover of saliva-producing acinar cells ranges from 50 to 125 days, gustatory and saliva secretory dysfunctions possibly persist after recovery from COVID-19. However, the direct interaction between SARS-CoV-2 and ACE2 in oral tissues to specifically induce oral symptoms mechanistically remains unclear.

It is well known that zinc modulates cell proliferation and differentiation, regulates inflammatory responses, and exhibits antiviral activity. In vivo zinc levels are an index of antiviral immunity and susceptibility to infectious and inflammatory diseases. Zinc deficiency predisposes people to infectious and inflammatory diseases and increases the production of pro-inflammatory cytokines: interleukin-6, interleukin-8, and tumor necrosis factor [128]. Zinc deficiency is associated with COVID-19 risk factors such as aging, malnutrition, medications, diabetes mellitus, and cardiovascular diseases. Zinc is present at high concentrations in taste bud membranes and salivary gland epithelial and myoepithelial cells [129,130]. It is considered that zinc is responsible for both taste perception and saliva secretion as zinc deficiency results in the reduction of taste sensitivity and impairment of saliva secretion [131]. Serum zinc levels are also related to the expression and activity of zinc-metalloenzyme ACE2 in patients with COVID-19 [132]. If cellular zinc concentrations are influenced by SARS-CoV-2 infection, the pathogenic mechanism underlying gustatory and saliva secretory dysfunctions could be speculated by zinc dynamics in COVID-19. Therefore, changes in in vivo zinc levels were verified by those of serum zinc concentrations.

Although a blood zinc level does not necessarily reflect the zinc status of an individual, zinc serum (or plasma) concentrations can be used as an indicator for the populational zinc status [133]. Table 1 summarizes serum zinc concentrations of COVID-19 patients reported by 15 studies [134,135,136,137,138,139,140,141,142,143,144,145,146,147,148], together with comorbidities such as diabetes mellitus, hypertension, respiratory disease, and cardiovascular disease that potentially induce zinc deficiency. Serum zinc concentrations ranged from 73 μg/dL to 106 μg/dL in controls or healthy subjects [135,136,141,145,146,148], whereas they ranged from 57 μg/dL to 80 μg/dL in COVID-19 patients [135,136,139,140,141,142,143,145,146,147,148]. The ratio of serum zinc deficiency was 57.4% in Indian COVID-19 patients [135] and 95.7% in Belgium COVID-19 patients [146] when the cutoff zinc concentration was defined as <80 μg/dL. Serum zinc concentrations are influenced by malnutrition, aging, pregnancy, iatrogenic diseases, and chronic diseases (diabetes mellitus, gastrointestinal disease, renal failure, etc.) [149]. Comorbidities were identified in a certain number of COVID-19 patients subjected to serum zinc analysis as shown in Table 1.

Figure 5A shows the quantitative results of zinc concentrations in serum obtained from COVID-19 patients and healthy subjects (or controls) in different cohorts [135,136,137,141,145,146,148]. Serum zinc concentration (mean ± SD) was 64.8 ± 7.5 μg/dL and 88.3 ± 14.0 μg/dL for COVID-19 patients and controls, respectively, indicating a significant difference between the two groups (*p* < 0.01) (Figure 5B).

An assessment of a Japanese cohort in the early phase of COVID-19 revealed that 85.7% of the patients have serum zinc concentrations below the cutoff value of < 70 μg/dL, which were markedly lower in patients with the severe disease than in patients with mild/moderate disease (*p* = 0.005) [134]. At hospital admission, Indian COVID-19 patients showed a median serum zinc concentration of 74.5 μg/dL that was significantly lower than control value of 105.8 μg/dL (*p* < 0.001) [135]. In a German cohort, serum zinc concentration was 71.7 ± 24.6 μg/dL and 97.6 ± 29.4 μg/dL for COVID-19 patients and healthy subjects, respectively, which were significantly different (*p* < 0.0001), and the ratio of zinc deficiency in non-survivors (dead) was higher than that in survivors discharged from hospital (73.5% vs. 40.9%, *p* < 0.0001) [136]. When serum zinc of Spanish COVID-19 patients was measured within the first 24 h of hospital admission, 74.2% of the patients showed lower concentration (63.5 ± 13.5 μg/dL) compared with normal subjects (> 84 µg/dL) [144]. Serum zinc concentrations of Iranian COVID-19 patients (68.4 ± 14.3 μg/dL for females and 66.7 ± 16.2 μg/dL for males) were significantly lower than controls (86.7 ± 11.8 μg/dL) (*p* < 0.001 for both) [141]. Hypozincemia at hospital admission was significantly associated with worse clinical presentation and higher mortality of Spanish COVID-19 patients [140]. Serum zinc concentrations in an Iranian COVID-19 cohort were significantly lower in the patients who died (94.2 ± 26.0 µg/dL) than in the patients who were admitted to the intensive care unit (ICU) (98.8 ± 30.5 µg/dL) or non-ICU and survived (118.8 ± 34.4 µg/dL) (*p* = 0.002 for both) [143]. Turkish COVID-19 patients had lower serum zinc concentrations depending on severity of the disease: asymptomatic (64.9 ± 12.4 µg/dL), mild (60.1 ± 18.1 µg/dL), moderate (56.9 ± 22.1 µg/dL), and severe (56.5 ± 18.1 µg/dL) compared with healthy subjects (87.3 ± 33.5 µg/dL) [148]. Fromonot et al. [150] compared plasma zinc levels between 152 COVID-19 patients and 88 non-COVID-19 patients who were admitted to French hospitals. They demonstrated that the prevalence of hypozincemia is significantly higher in COVID-19 patients than in non-COVID-19 patients (*p* = 0.003). In order to investigate time-dependent changes in zinc levels, Yasui et al. [134] performed a follow-up study of Japanese COVID-19 patients who were admitted to the ICU, treated with enteral nutrition delivered from the tube inserted through the nose, and finally discharged from the hospital. Serum zinc concentrations of the patients were below or near the zinc deficiency cutoff concentration of 70 µg/dL for 4 weeks after disease onset. From these results, COVID-19 patients are characterized by hypozincemia, in which serum zinc concentrations are decreased depending on COVID-19 severity and maintain a relatively low level for a certain period after recovery from COVID-19.

## 4. Discussion

The pathogenesis of gustatory and saliva secretory dysfunctions cooccurring in COVID-19 patients is hypothetically speculated by focusing on zinc dynamics, zinc-metalloenzymes, metallothioneins, and zinc transporters.

### 4.1. Pathogenic Speculation from the Perspective of Zinc Deficiency

Zinc per se possesses the ability to inhibit the replication and growth of SARS-CoV-2 by acting on the viral RNA-dependent RNA polymerase. Decreasing zinc levels is favorable for the interaction of viral spike proteins with cellular receptor ACE2, whereas increasing zinc levels inhibits the expression of ACE2 [151]. Zinc deficiency results in the reduction of the host immunity, thereby increasing susceptibility to SARS-CoV-2 infection [152]. While blood zinc concentrations respond to infection and inflammation, hypozincemia associated with viral infection is improved by treatment with antiviral agents [153]. Despite antiviral and immune-enhancing effects of zinc, Singh et al. [154] revealed that zinc-sufficient status is positively correlated with COVID-19 mortality in European populations. Therefore, COVID-19 oral symptoms can be pathogenically discussed by relating to zinc deficiency resulting from SARS-CoV-2 infection, not to zinc deficiency as the pathogenic trigger causing COVID-19.

Zinc-deficient patients complain of taste disorders simultaneously with xerostomia and exhibit morphological changes in parotid and submandibular glands [155]. Zinc is richly present in taste bud cells and salivary gland cells. If SARS-CoV-2 infection induces hypozincemia by redistributing zinc from blood to and accumulating zinc in the liver at the expense of zinc in other tissues as found in septic patients [138], the resultant intracellular zinc deficiency may make an impact on taste perception and saliva secretion in COVID-19 patients.

Arnaud et al. [156] analyzed blood zinc levels in a large sample cohort of French adults (7448 females, 35–65 years old and 4926 males, 45–65 years old). Serum zinc concentrations showed a significant geographical difference, gender difference, and negative association with age. Cediel et al. [157] systematically reviewed zinc deficiency in Latin America and the Caribbean. The prevalence of low serum zinc concentrations in women and children appeared to vary among different countries: Mexico, Guatemala, Colombia, and Ecuador. When Wu et al. [158] assessed different ethnic populations, African Americans with heart failure were found to have more antioxidant zinc deficiency than Whites. Hennigar et al. [133] evaluated serum zinc concentrations in the US population consisting of males (*n* = 2193) and females (*n* = 2154) aged ≥ 6 years. They revealed that overall serum zinc concentrations are higher in males than in females (84.9 ± 0.8 μg/dL vs. 80.6 ± 0.6 μg/dL, *p* < 0.0001). A Japanese cohort showed that the proportion of serum zinc deficiency is larger in females than in males [159]. In a comparative study of Hennigar et al. [133], serum zinc concentrations were decreased with age in the US population. Persistent zinc deficiency is common in the elderly and related to their higher susceptibility to infectious diseases. Taken together, serum zinc concentrations are considered to vary depending on differences in ethnicity, gender, and age. Such intrinsic zinc status may characterize zinc deficiency of COVID-19 patients with gustatory and saliva secretory dysfunctions in addition to the consequence resulting from SARS-CoV-2 infection.

### 4.2. Zinc and Zinc-Metalloenzyme Carbonic Anhydrase

Zinc is essential not only for the gustatory function at a level of taste buds and taste stimulus-transmitting nerves but also for the regeneration and maintenance of taste cells [128]. The appropriate concentration of cellular zinc is pivotal to maintaining the functional and morphological normality of cells. Intracellular zinc deficiency exhibits adverse effects on rat vallate papillae to change the number and size of taste buds [160]. In vallate papillae of zinc-deficient rats, the number of bitter taste receptor gene TAS2R-positive cells is markedly smaller compared with normal rats as well as the number of salty taste-mediating epithelial sodium channel ENaC-positive cells [161]. Zinc is also localized on the membrane surfaces, granules, and vesicles of the glandular epithelial cells and in the pits of the myoepithelial cells in rat submandibular glands together with zinc-metalloenzymes; therefore zinc is considered to physiologically participate in saliva secretion [162].

Taste recognition is involved in various effects of saliva and taste alteration is associated with the concentration change of salivary components [163]. Viral infection and inflammation of salivary glands result in salivary compositional changes. Abduljabbar et al. [164] raised the question “Does SARS-CoV-2 infection alter the salivary components and their composition to induce early COVID-19 symptoms of ageusia and hypogeusia?” because taste perception is dependent on the flow rate of saliva and salivary constituents including zinc and zinc-binding protein. Zinc-metalloenzyme carbonic anhydrases bind to cellular zinc with high affinity and their activity is dependent on the presence of zinc. Carbonic anhydrases catalyze the hydration of carbon dioxide to bicarbonate and a proton, regulating many physiological processes such as ion transport, pH regulation, and fluid balance. In addition, they play an important role in the production and secretion of saliva and the regulation of saliva pH. Human carbonic anhydrases have 15 isoforms and their dysregulated expression is related to various diseases as carbonic anhydrase I, IV, IX, and XII isoforms are abnormally expressed in diseased conditions such as rheumatoid arthritis, cerebral ischemia, and cancers [165].

Among different isozymes, carbonic anhydrase VI was found to be localized in rat taste buds and salivary glands and also in human parotid and submandibular glands [166]. Carbonic anhydrase VI (previously identified as “gustin”) is secreted into saliva by the serous acinar cells of the mammalian parotid and submandibular glands [163]. Since carbonic anhydrase VI acts as a trophic factor to promote the growth, development, and maintenance of taste buds and fungiform taste papillae, this zinc-metalloenzyme is linked to gustatory function as it influences taste sensitivity to bitter tastant 6-*n*-propylthiouracil [163]. Henkin et al. [167] measured carbonic anhydrase VI and zinc in parotid saliva of patients who developed hypogeusia after infection with influenza virus and morphologically examined circumvallate papillae biopsied from the patients. Concentrations of both zinc and carbonic anhydrase VI in parotid saliva were much lower in the patients than in healthy subjects and they were more significantly decreased compared with serum zinc concentrations. Taste buds in the circumvallate papillae of the patients exhibited pathological changes such as vacuolization and cellular degeneration. Their results suggest that a decrease of carbonic anhydrase VI is associated with the occurrence of taste bud abnormalities and the subsequently induced gustatory dysfunction. In a following study [168], they verified the effect of oral zinc treatment (100 mg zinc daily) on patients with complaints of gustatory dysfunction. After the treatment for 4–6 months, the patients showed significant increases of carbonic anhydrase VI and zinc in parotid saliva together with the improvement of taste, and the taste buds returned to the morphologically normal state. They speculated that carbonic anhydrase VI could promote the growth and development of taste buds through its effect on taste bud stem cells. Dysgeusia was reported to occur as a side effect of carbonic anhydrase inhibitors used for the treatment of glaucoma and idiopathic dysgeusia was effectively treated with zinc gluconate [166]. Goto et al. [169] investigated the effects of zinc deficiency on carbonic anhydrase activity in the tongue epithelia and submandibular glands of rats that were given free access to zinc-deficient, low-zinc, or zinc-sufficient diets for 6 weeks. They revealed that zinc deficiency significantly reduces the carbonic anhydrase activity with the degree correlating to dietary zinc levels. In their enzyme histochemical experiment, zinc-deficient rats showed a weaker reactivity to carbonic anhydrase in taste buds of the circumvallate papillae compared with zinc-sufficient rats. When nutrient intake and taste perception were comparatively evaluated, a significant number of subjects had insensitive taste (sweet, salty, sour, or bitter taste) in association with the low intake of zinc [170].

Zinc is redistributed from blood to the liver during infection-induced systemic inflammation like sepsis [138,171], resulting in a decrease in serum zinc concentrations. Viral infection and inflammation lead to hepatic zinc accumulation at the expense of zinc in other tissues, reducing the tissue zinc levels. If a decrease of zinc concentrations in oral tissues is conceivable as a result of SARS-CoV-2 infection [172], COVID-19 may induce intracellular zinc deficiency and inhibit the activity of carbonic anhydrase localized in taste bud cells and salivary gland cells with the resultant negative effects on taste perception and saliva secretion. In zinc-deficient model experiments of Ishii et al. [162], rats were fed zinc-deficient diets or administered with zinc chelator dithizone. Experimentally induced chronic and acute zinc deficiency showed a significant decrease of saliva secretion from submandibular glands with morphological changes.

The genotypes of genetic polymorphism may be related to different expression levels of carbonic anhydrase VI between East Asian and European populations as found in an ethnic or racial difference of ACE2 expression [173]. Carbonic anhydrase VI exhibits seven single nucleotide polymorphisms, which have been associated with changes in saliva property and dental caries susceptibility because salivary carbonic anhydrase VI is implicated in gustatory dysfunction and also in dental caries occurrence [174,175]. Polymorphism in the carbonic anhydrase VI gene may contribute to geographically different characteristics of the early gustatory dysfunction of COVID-19 patients and the gustatory sequelae of COVID-19 survivors.

ACE2 as one of the renin-angiotensin system components is present not only in fungiform papillae taste cells but also in the acinar, duct, and myoepithelial cells of parotid, submandibular, and sublingual glands as well as carbonic anhydrase VI. ACE2 degrades angiotensin II to angiotensin (1–7). Similar to ACE2, neprilysin (membrane metallo-endopeptidase) is able to produce angiotensin (1–7) by cleaving angiotensin I and angiotensin (1–9). Zolfaghari Emameh et al. [176] defined the significance of co-expression of three zinc-metalloenzymes: ACE2, neprilysin, and carbonic anhydrase in the pathogenesis of COVID-19.

### 4.3. Cellular Zinc Homeostasis

In mammalian cells, metalloproteins or metalloenzymes tightly bind to zinc. In addition to them, metallothioneins and zinc transporters also have a high binding affinity for zinc [130]. Metallothioneins control cellular zinc homeostasis cooperatively with zinc transporters [177]. The expression of metallothioneins and zinc transporters can be a biomarker of zinc status [178].

Metallothioneins are cysteine-rich metal-binding proteins that function as a zinc acceptor or a zinc donor to mediate the movement of cellular zinc to other zinc-binding proteins. The human metallothionein gene family consists of at least 18 isoforms that are divided into 4 classes: metallothionein-1, metallothionein-2, metallothionein-3, and metallothionein-4. They physiologically contribute to metal detoxification, homeostatic regulation of metals, protection against oxidative stress, and neuroprotection. Metallothionein-1 and metallothionein-2 are ubiquitously present in many cell types of tissues, particularly in the liver and kidney at high concentrations. In contrast to metallothionein-1/2, it has been considered that metallothionein-3 and metallothionein-4 are localized in the central nervous system and stratified squamous epithelia, respectively. Interestingly, metallothionein-3 was recently found to be expressed in tissues other than the brain: taste buds [179] and salivary glands [180]. Its expression specific to oral tissues suggests potential effects of metallothionein-3 on taste perception and saliva secretion.

Zinc is compartmentalized into or out of intracellular organelles and vesicles by zinc transporters. Zinc transporters are divided into two distinct families. The Zrt- and Irt-like protein (ZIP) family of zinc importers and the Zn transporter (ZnT) family of zinc exporters have opposite roles in controlling cellular zinc homeostasis. The ZIP family consists of 14 members (ZIP1 to ZIP14) that facilitate zinc influx into the cytosol or increase cytoplasmic zinc by transporting zinc from the extracellular space or intracellular organelles to the cytosol. The ZnT family consists of nine members (ZnT1 to ZnT8 and ZnT10) that facilitate zinc efflux from the cytosol or decrease cytoplasmic zinc by transporting cytosolic zinc to the extracellular space or intracellular organelles. Coordinated zinc mobilization by these zinc transporters contributes to different physiological functions [177]. Yang et al. [181] found that ZIPs and ZnTs are highly expressed in human salivary glands, suggesting the possibility that zinc transporters may have some effect on saliva secretion.

### 4.4. Metallothionein

In hypozincemia caused by viral and bacterial infection, a decrease of serum zinc concentrations is due to the redistribution of zinc from blood to the liver, which is induced by the expression of hepatic metallothionein at the acute stage of infectious diseases [138,171]. Ghoshal et al. [182] reported that viral infection induces metallothionein expression in mouse liver and lung. Serum zinc concentrations are inversely related to intestinal metallothionein levels [183]. In a recent study by Livanos et al. [184], human intestinal tissues biopsied from COVID-19 patients showed that the genes of metallothionein-1/2 are upregulated on lamina propria and epithelial compartments.

Of metallothioneins, metallothionein-3 is specifically localized in oral tissues participating in taste perception and saliva secretion. Hozumi et al. [179] conducted RT-PCR, Western blot, and immunohistochemical analyses to determine metallothionein-3 mRNA and its protein in various peripheral tissues of rats. They revealed that metallothionein-3 is expressed in taste bud cells in the tongue. Irie et al. [180] immunohistochemically examined autopsy samples of human submandibular and sublingual glands. Their results indicated that metallothionein-3 is expressed in the duct and acinar cells of large salivary glands. If the expression of metallothionein-3 is upregulated in taste bud cells and salivary gland cells to compensate for intracellular zinc deficiency induced by COVID-19 as observed in hepatic and intestinal cells, increasing cellular metallothionein-3 may have some impact on taste perception and saliva secretion.

Metallothionein-3 concentration-dependently exerts biphasic effects on neural cells to inhibit their growth at relatively high concentrations and advance at relatively low concentrations [180,185]. As mentioned by Lee and Koh [186], metallothionein-3 inhibits neurite outgrowth and promotes neuronal death likely by releasing cytotoxic zinc in addition to functioning as a metal detoxicant like other metallothioneins. Metallothionein-3 has been referred to as an inhibitor of neurite formation in developing neurons and postinjury regenerative neurite sprouting [187]. It has been suggested that the formation and development of taste buds and salivary glands depend on afferent nerves and sensory neurite growth [188]. If the expression of metallothionein-3 is upregulated by SARS-CoV-2 infection, the concentration change of cellular metallothionein-3 may affect taste buds and salivary glands to impair their gustatory and secretory functions. Given the cell turnover of taste buds and salivary gland acini to be weeks to months, gustatory and saliva secretory dysfunctions could persist after recovery from COVID-19.

Scudiero et al. [189] determined the expression profiles of metallothionein isoforms in the cerebral cortex and hippocampus of adolescent (2 months), adult (4 and 8 months), and middle-aged (16 months) rats by real-time PCR analysis. They showed the age-dependent metallothionein gene expression that metallothionein-3 transcripts are significantly increased at 16 months in both cortical and hippocampal areas, but decreased at 2–8 months. The influence of intracellular zinc deficiency should be more significant on the tissues where metallothionein-3 is increasingly expressed and localized. If metallothionein-3 is age-dependently expressed in taste buds and salivary glands similarly to the brain, such an expression pattern may be associated with the age-dependent characteristics of oral symptoms in COVID-19 patients.

Dysregulated metallothionein expression is found in adenoid cystic carcinoma of human salivary glands [190,191], in which metallothionein expression is increased to activate transcriptional factors by donating zinc [192]. Zinc and cytokines induce metallothioneins in the brain and liver, whereas interleukin-6 deficiency decreases the expression of metallothioneins in the central nervous system [193]. While SARS-CoV-2 causes sialadenitis of submandibular salivary glands and inflammation of parotid salivary glands (parotitis), COVID-19 survivors frequently develop salivary gland ectasia, exhibiting the hyperinflammatory response to SARS-CoV-2 infection. Mammalian taste bud cells express several pro-inflammatory cytokines and inflammation affects taste buds through cytokine signaling pathways by attenuating cell proliferation and interfering with taste cell renewal [194]. Pro-inflammatory cytokines including interleukin-1 and interleukin-6 induce the gene expression of metallothioneins [183].

### 4.5. Zinc Transporter

Hypozincemia induced by inflammation and infection is attributable to zinc redistribution, which is promoted by ZIP14 induction in hepatocytes [171]. Increased pro-inflammatory cytokines also cause the redistribution of zinc through up-regulation of ZnTs and ZIPs [136]. Interleukin-6 not only regulates hepatic ZIP14 but also contributes to hypozincemia of the acute-phase response to infection and inflammation [195]. Yang et al. [181] examined the expression of 14 ZIP and 10 ZnT transporters in human organs and tissues by RT-PCR and immunohistochemistry. Their results indicated that human zinc transporters show a tissue-specific expression pattern, that is, ZIP6, ZIP7, ZIP8, ZIP9, and ZIP11 are highly expressed in salivary glands, followed by ZIP2, ZIP3, ZIP13, and ZIP14 at medium levels, and ZnT5, ZnT7, and ZnT9 at moderate levels. Dysregulation of zinc transporter expression is observed in various diseases as ZnTs and ZIPs are downregulated and upregulated, respectively, in pancreatic cancers and mammary tumors [181,196,197]. Although the relation of ZIPs and ZnTs to saliva secretory and gustatory functions still needs to be elucidated, if the expression of zinc transporters is dysregulated in association with COVID-19-induced intracellular zinc deficiency, cellular zinc homeostasis could be disturbed, affecting the activity of zinc-metalloenzymes (carbonic anhydrase and ACE2) in taste buds and salivary glands with the subsequent functional declines of taste perception and saliva secretion.

### 4.6. Zinc Supplementation

Complication rate, acute respiratory distress syndrome frequency, hospital stay, and mortality are increased in zinc-deficient COVID-19 patients. Zinc supplementation has the potential to prevent and treat COVID-19 because of immune-boosting and SARS-CoV-2 replication-suppressing effects. Since zinc deficiency is closely associated with gustatory and saliva secretory dysfunctions, zinc supplementation is also expected to improve oral symptoms in COVID-19 patients and survivors. Oral administration of zinc would lead to an increase of intracellular zinc concentrations with the subsequent enhancement of carbonic anhydrase activity and influence on metallothionein and zinc transporter expression. Aydemir et al. [198] evaluated the response of zinc-binding proteins to a supplement of zinc (15 mg per day) by measuring transcript abundance in monocytes, T lymphocytes, and granulocytes prepared from peripheral blood of human subjects using quantitative real-time RT-PCR. A modest dietary zinc supplementation increased the gene expression of metallothionein-1/2 and changed transcript levels for the zinc transporter genes (increased ZnT1 and decreased ZIP3) in leukocyte populations.

Carlucci et al. [199] provided the first in vivo evidence for the efficiency of zinc supplementation in COVID-19. In their study, COVID-19 patients (*n* = 411) were administered with zinc sulfate (220 mg capsule containing 50 mg elemental zinc twice daily for 5 days) combined with zinc ionophore hydroxychloroquine (400 mg load followed by 200 mg twice daily for 5 days) and azithromycin (500 mg once daily). When comparing with patients taking hydroxychloroquine and azithromycin alone (*n* = 521), zinc supplementation was effective in improving outcomes of hospitalized COVID-19 patients. In the literature, few studies (ongoing or proposed clinical trials) have assessed the effect of zinc supplementation on oral symptoms associated with COVID-19. However, non-COVID-19 cases may give important implications for the application of zinc supplementation to COVID-19 patients with gustatory and saliva secretory dysfunctions. In a randomized clinical trial for patients with idiopathic dysgeusia, oral administration of zinc gluconate (140 mg/day for 3 months) improved gustatory function and reduced the severity of dysgeusia [200]. By supplementing with elemental zinc at 68 mg/day (300 mg/day as a zinc L-carnosine complex, Polaprezinc) for 12 weeks, taste sensitivity was significantly enhanced in patients suffering from gustatory dysfunction [201]. Oral administration of zinc acetate (15 mg zinc/day for 5 weeks) also increased the flow rate of stimulated saliva from parotid glands of adult subjects [202]. When oral zinc sulfate (300 mg/day for 6 months) was prescribed for patients complaining of oral symptoms, xerostomia and taste disorder was relieved or improved in 57.9% and 72.7% of the patients, respectively [155]. With respect to COVID-19 neurological symptoms, there is one clinical trial that may encourage zinc supplementation as the therapy of COVID-19 oral symptoms. Abdelmaksoud et al. [139] evaluated the recovery of chemosensory disorders of patients (*n* = 105) with mild to critical COVID-19 who received zinc therapy (220 mg zinc sulfate equivocal to 50 mg elemental zinc twice daily). When the patients were followed up until the results of RT-PCR test were negative, their recovery time of gustatory and/or olfactory function was significantly shorter compared with controls.

Zinc supplementation for COVID-19 patients was comprehensively reviewed by Joachimiak [203], Rahman and Idid [204], and Chinni et al. [205]. Of particular interest is an excellent review recently reported by Santos [172] about the therapeutic zinc supplementation for COVID-19 ageusia and its rationale.

## 5. Conclusions

Gustatory dysfunction (ageusia, hypogeusia, and dysgeusia) and saliva secretory dysfunction (xerostomia, dry mouth, and hyposalivation) frequently cooccur in patients infected with SARS-CoV-2, and both dysfunctions can persist after recovery from COVID-19. These cooccurring oral symptoms could be interpreted by in vivo zinc level dynamics of COVID-19 patients because zinc, zinc-metalloenzymes, metallothioneins, and zinc transporters are expressed and localized in oral tissues participating in taste perception and saliva secretion.

Hypozincemia is highly prevalent in COVID-19 patients. A decrease in serum zinc concentrations should reflect an alteration of cellular zinc levels in peripheral tissues. If SARS-CoV-2 infection induces intracellular zinc deficiency in oral tissues, the activity of zinc-metalloenzyme carbonic anhydrase VI localized in taste bud cells and salivary gland cells may be reduced to adversely affect taste perception and saliva secretion. The expression of zinc-binding proteins to control zinc homeostasis is referred to as a biomarker of zinc status. Metallothionein-3 and zinc transporters are expressed in taste buds and salivary glands. If the expression of metallothionein-3 is upregulated to compensate for intracellular zinc deficiency induced by COVID-19, its inhibitory effect on neurite formation may influence gustatory and saliva secretory functions. ZIPs and ZnTs expression in salivary glands may be dysregulated by SARS-CoV-2 infection to disturb cellular zinc homeostasis. Zinc deficiency associated with COVID-19 oral symptoms supports the use of zinc supplementation for improving them.

Gustatory dysfunction is highly prevalent in COVID-19 patients infected with early SARS-CoV-2 strains, the wild type and the Delta variant (B.1.617.2). Since the first discovery and classification as a variant of concern, the Omicron variant (B.1.1.529) with greater transmissibility has globally spread very quickly to be the dominant strain worldwide as of February 2022. There is increasing evidence that most cases of Omicron infection are asymptomatic or mildly severe, whereas anecdotal reports suggest that patients infected with the Omicron variant less commonly have gustatory dysfunction. Serum zinc concentrations are significantly lower in patients with severe COVID-19 than in patients with asymptomatic and mild to moderate COVID-19. Some pathophysiological studies indicated that the decreasing degree of serum zinc concentrations is dependent on the severity of COVID-19. It is of much interest to characterize oral symptoms in patients infected with the Omicron variant from the point of view of zinc dynamics.

## Figures and Tables

**Figure 1 life-12-00353-f001:**
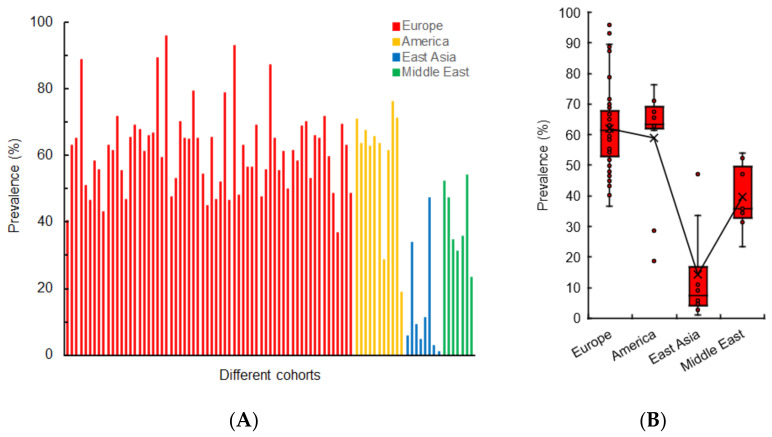
Prevalence of gustatory dysfunction in the early phase of COVID-19 (**A**) and geographical comparison (**B**).

**Figure 2 life-12-00353-f002:**
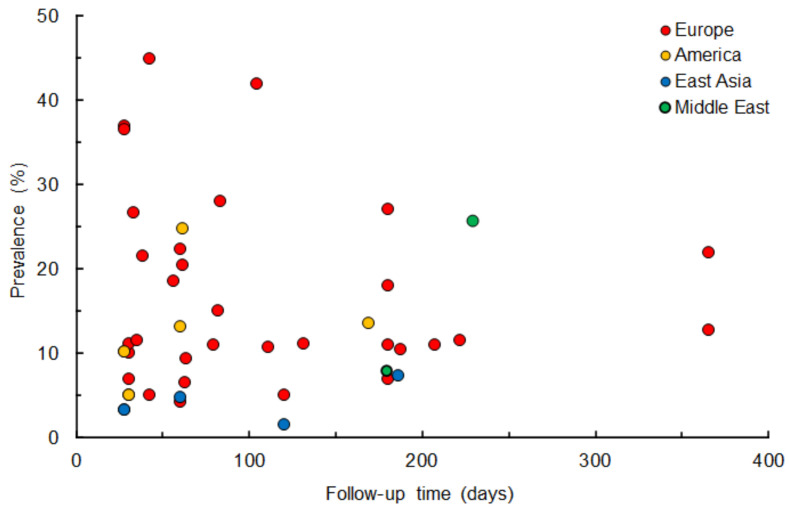
Prevalence of persistent gustatory dysfunction in COVID-19 survivors followed up after symptom onset, hospital admission, or disease diagnosis in geographically different cohorts.

**Figure 3 life-12-00353-f003:**
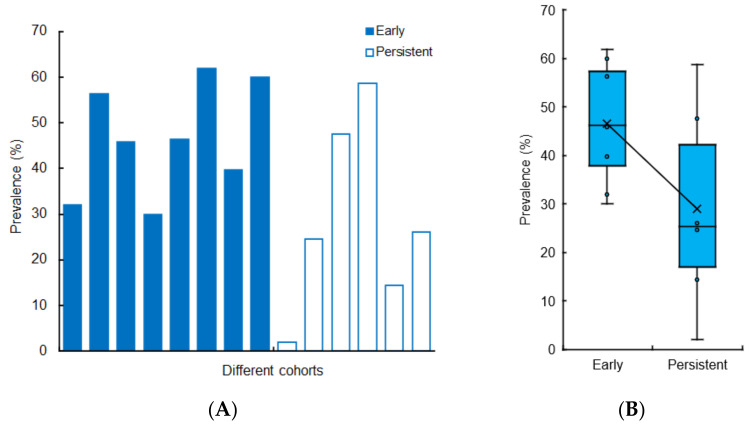
Prevalence of early and persistent saliva secretory dysfunctions in different COVID-19 cohorts (**A**) and their comparison (**B**).

**Figure 4 life-12-00353-f004:**
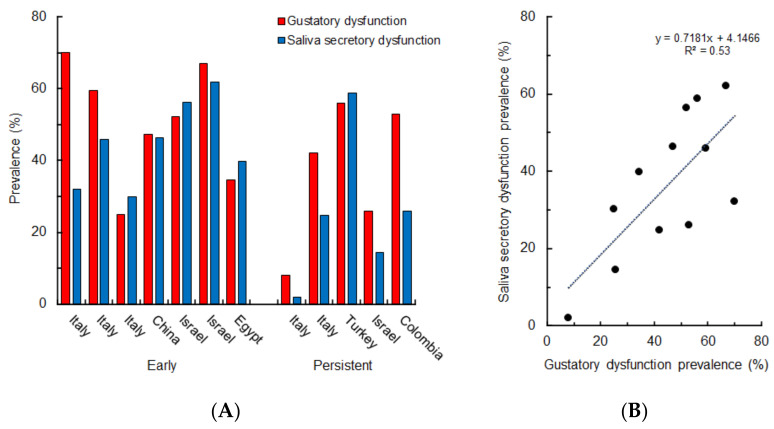
Prevalence of both gustatory dysfunction and saliva secretory dysfunction in the early phase of COVID-19 and persistent after recovery from COVID-19 in different cohorts (**A**) and the relation between gustatory dysfunction and saliva secretory dysfunction (**B**).

**Figure 5 life-12-00353-f005:**
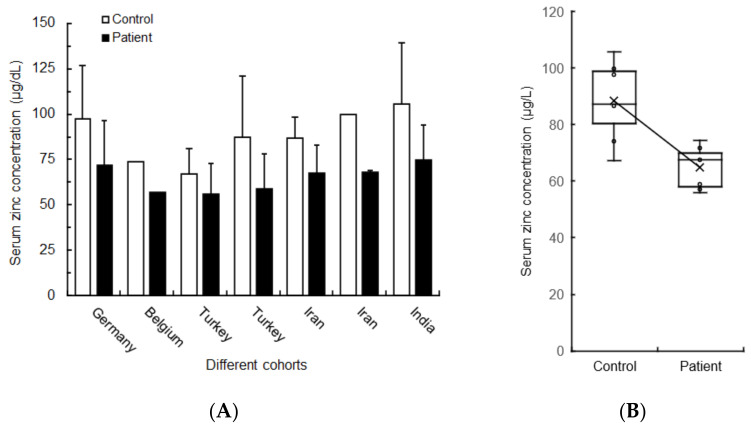
Serum zinc concentrations of COVID-19 patients and controls in different cohorts (**A**) and comparison between them (**B**).

**Table 1 life-12-00353-t001:** Serum zinc concentrations and zinc deficiency in COVID-19 patients.

Subjects	Disease Severity	Country of Subjects	Number of Subjects	Mean or Median Age (Year, Range)	Female (%)	Test Samples	Mean or Median Concentration (μg/dL, ± SD or Range)	Deficiency Ratio (%) Deficiency Defined at	Comorbidities	Reference
Hospitalized patients	Severe: 24.1% Mild and moderate: 75.9%	Japan	29	≥65 years Severe: 57.1% Mild and moderate: 31.8%	Severe: 42.9 Mild and moderate: 45.5	Collected the first day of hospitalization and 2–3 days later	Severe: 62.4 ± 19.2 Mild and moderate: 87.7 ± 19.1	Severe: 85.7 Mild and moderate: 13.6 <70 μg/dL	Total HT (41.4%) DM (20.7%) KD (10.3%) RD (6.9%) CD (6.9%)	Yasui et al. [134]
Hospitalized patients diagnosed by RT-PCR test	Mild: 83.0% Moderate: 6.4% Severe: 10.6%	India	Patients: 47 Control: 45	34 (18–77) 32 (18–60)	38.5 Control: 32.3	Collected 6 h after hospital admission	Patients: 74.5 (53.4–94.6) Control: 105.8 (95.7–120.9)	57.4 <80 μg/dL	DM (14.8%) HT (14.8%) CD (3.7%) TD (3.7%)	Jothimani et al. [135]
Hospitalized patients diagnosed by RT-PCR test	Survivors: 82.9% Non-survivors: 17.1%	Germany	35	Total: 77 (38–94) Survivors: 70 (38–91) Non-survivors: 89 (81–94)	54.3 51.7 66.7	Collected consecutively	Patients: 71.7 ± 24.6 Healthy subjects: 97.6 ± 29.4	Survivors: 40.9 Non-survivors: 73.5 <63.87 μg/dL	NR	Heller et al. [136]
Hospitalized pregnant women diagnosed by RT-PCR test	NR (in the first, second, and third trimesters of pregnancy)	Turkey	Patients: 100 Control: 100	Patients in the first trimester: 28 (17–38) Patients in the second trimester: 29 (18–41) Patients in the third trimester: 30 (22–41)	100	Measured on admission to hospital	In the first trimester Patients: 56.0 ± 16.6 Control: 67.2 ± 13.9 In the second trimester Patients: 46.4 ± 12.7 Control: 52.8 ± 12.6 In the third trimester Patients: 46.8 ± 12.5 Control: 54.4 ± 13.6	NR	NR	Anuk et al. [137]
Hospitalized patients diagnosed by RT-PCR test	Admitted to ICU	Brazil	269	74 (66–81)	48.7	Measured at admission to ICU	Patients: 59.8 (49.7–67.7)	79.6 <70 μg/dL	HT (74.0%) DM (42.4%) PD (27.9%) CD (27.5%) KD (13.0%)	Gonçalves et al. [138]
Patients diagnosed by RT-PCR test	Mild: 33.6% Moderate: 42.5% Severe: 15.7% Critical 8.2%	Egypt	Total: 134 Mild: 45 Moderate: 57 Severe: 21 Critical: 11	Mild: 31.8 Moderate: 47.8 Severe: 59.1 Critical: 69.5	Mild: 46.7 Moderate: 42.1 Severe: 28.6 Critical: 45.5	Collected prior to zinc therapy	Mild: 67 ± 18 Moderate: 62 ± 14 Severe: 73 ± 18 Critical: 72 ± 22	NR	Total: DM (~27.3%) HT (~23.8%) CD (~9.1%)	Abdelmaksoud et al. [139]
Hospitalized patients diagnosed by RT-PCR test	NR	Spain	249	65 (54–75)	49.0	Measured at hospital admission	Patients: 61	23.3 <50 μg/dL	HT (56.6%) KD (28.1%) DM (21.3%) CD (14.9%) PD (8.8%)	Vogel-González et al. [140]
Hospitalized patients diagnosed by RT-PCR test	Severe: 39.8%	Iran	Patients: 93 Control: 186	51 (40–61)	55.9	NR	Total patients: 67.6 ± 15.1 Female patients: 68.4 ± 14.3 Male patients: 66.7 ± 16.2 Total control: 86.7 ± 11.8 Female control: 79.4 ± 10.9 Male control: 86.6 ± 14.0	52.7 NR	CD (21.5%) DM (16.1%) HT (10.8%) PD (8.6%)	Elham et al. [141]
Patients diagnosed by RT-PCR test and admitted to ICU	Minimal: 15.0% Mild: 28.3% Moderate: 45.0% Severe: 11.7%	Iran	Total: 60 APACHE score < 25: 40 APACHE score ≥ 25: 20	Total: 53.5 APACHE score < 25: 50.0 APACHE score ≥ 25: 56.0	Total: 35.0 APACHE score < 25: 40.0 APACHE score ≥ 25: 25.0	NR	Total patients: 70.0 ± 44.5 APACHE score < 25: 80.0 ± 32.8 APACHE score ≥ 25: 50.5 ± 18.0	NR	APACHE score < 25: DM and HT (27.5%) DM (12.5%) TD (15.0%)	Beigmohammadi et al. [142]
Hospitalized patients diagnosed by RT-PCR test	Admitted to non-ICU and survived Admitted to ICU Died	Iran	Total: 293	53 (38–65)	50.2	Measured within 3 days of admission	Admitted to non-ICU and survived: 118.8 ± 34.4 Admitted to ICU: 98.8 ± 30.5 Died: 94.2 ± 26.0	NR	CD (27.6%) DM (16.0%) HT (6.1%) KD (6.1%)	Shakeri et al. [143]
Hospitalized patients diagnosed by RT-PCR test	Moderate: 32.5% Severe: 67.5%	Spain	120	58.7	35.8	Assessed within the first 24 h of hospital admission	63.5 ± 13.5	74.2 <84 μg/dL	HT (32.5%) DM (16.7%) PD (5.0%)	Tomasa-Irriguible et al. [144]
Hospitalized patients diagnosed by RT-PCR test	Severe (admitted to ICU): 49.6% Non-severe (non-ICU ward): 50.6%	Iran	Total: 226 ICU: 112 Non-ICU: 114	Total: 56.426 ICU: 56 Non-ICU: 56.7	Total: 49.6 ICU: 50.0 Non-ICU: 49.2	NR	Normal females: 77.0–114 Normal males: 72.6–127 Total patients: 67.9 ± 1.1 Severe: 67.3 ± 1.8 Non-severe: 68.4 ± 1.4	NR	Total: KD (24.3%) DM (21.2%) CD (20.4%) PD (7.5%)	Bagher Pour et al. [145]
Hospitalized patients diagnosed by RT-PCR test	NR	Belgium	139	65 (54–77)	34.5	Assessed within the first 72 h of hospital admission	Plasma zinc concentration Patients: 57 (45–67) Control: 74 (80–120)	95.7 <80 μg/dL	HT (64.7%) DM (36.0%) KD (22.3%) PD (18.0%) CD (16.5%)	Verschelden et al. [146]
Hospitalized patients diagnosed by RT-PCR test	Mild, Moderate, Severe, Critical, Died	Belgium	79	(18–100)	30.4	Analyzed at hospital admission	Total patients: 73.5 ± 16.6 Female patients: 74.8 ± 17.3 Male patients: 72.9 ± 16.4 with DM: 76.7 ± 17.1 with CD: 73.7 ± 12.7	70.6 <66 μg/dL	DM (30.4%) CD (27.8%)	Du Laing et al. [147]
Hospitalized patients diagnosed by RT-PCR test	Asymptomatic: 6.7% Mild: 25.0% Moderate: 46.7% Severe: 21.7%	Turkey	Patients: 60 Control: 32	Total patients: 45.5 Asymptomatic: 41.3 Mild: 31.9 Moderate: 54.1 Severe: 58 Control: 48.8	Total patients: 46.7 Asymptomatic: 50.0 Mild: 13.3 Moderate: 53.6 Severe: 69.2 Control: 46.7	Collected at hospital admission	Total patients: 58.8 ± 19.5 Asymptomatic: 64.9 ± 12.4 Mild: 60.1 ± 18.1 Moderate: 56.9 ± 22.1 Severe: 56.5 ± 18.1 Control: 87.3 ± 33.5	NR	NR	Kocak et al. [148]

NR: Not reported. RT-PCR: Reverse transcription-polymerase chain reaction. ICU: Intensive care unit. DM: Diabetes mellitus. HT: Hypertension. KD: Kidney disease. RD: Respiratory disease. CD: Cardiovascular disease. TD: Thyroid disease. PD: Pulmonary disease. APACHE: Acute physiologic assessment and chronic health evaluation. Demographic data and disease severity at baseline.

## Data Availability

Not applicable.

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
