# Peer review of "Gustatory and Saliva Secretory Dysfunctions in COVID-19 Patients with Zinc Deficiency"

_life, 2022, doi:10.3390/life12030353_

Round 1

Reviewer 1 Report

Dear Author,

The review is well done. To update our knowledge  about COVID-19 symptomatology and pathophisiology is a main topic during this time.

I just recommend some minor revision to better clarify some passages of your papar.

In the Introduction section please provide a description of another clinical manifestation: Anosmia. As reported in literature anosmia and ageusia are the early manifestation of the infection.

In material and methods please change "scientific articles" with "Literature analysis". I suggest also to split results section from the discussion in order to better understand the differences of the two section. Please clarify the pathogenesis from line 225 to 230. 

I strongly recommend a Moderate english revision by a native speaker in order to make more readable fluently your paper.

Author Response

Author’s Responses to the Comments of Reviewer 1

Comments and Suggestions for Authors:

The real significance of good analysed dates in this article is based on fundamental aspects of pathophysiological mechanisms of gustatory dysfunctions after covid and real practical recomendations which we can have after reading.  The work is well organized and logic.  I think it will be very interesting for many readers.

I just recommend some minor revision to better clarify some passages of your papar.

The author appreciates reviewer’s suggestive comments.

In the Introduction section please provide a description of another clinical manifestation: Anosmia. As reported in literature anosmia and ageusia are the early manifestation of the infection.

Descriptions about “anosmia” were added to Introduction and two references were newly cited (please see the revised MS: Line 30-34, References 3 & 4).

In material and methods please change "scientific articles" with "Literature analysis".

The meaning of “literature analysis” is slightly different from what the author wanted to say. So, the original two sentences containing “scientific articles” were revised (please see the revised MS: Line 62 in Materials and Methods, Line 10-11 in Abstract).

I suggest also to split results section from the discussion in order to better understand the differences of the two section.

According to the suggestion, the original “Results and Discussion” was split into two sections (3. Results and 4. Discussion). The section of “Results” focused on the overviews of gustatory dysfunction, saliva secretory dysfunction and zinc deficiency, and the section of “Discussion” on the pathogenic speculation (please see the revised MS: Line 78-321 for Results, Line 322-603 for Discussion).

Please clarify the pathogenesis from line 225 to 230.

The corresponding sentences were modified (please see the revised MS: Line 230-237).

I strongly recommend a Moderate english revision by a native speaker in order to make more readable fluently your paper.

The original manuscript was corrected throughout all pages after grammatical revision by a native English speaker.

Reviewer 2 Report

The real significance of good analysed dates  in this article is based on fundamental aspects of pathophysiological mechanisms of gustatory dysfunctions after covid  and real practical recomendations which we can have after reading.  The work  is well organized and logic.  I think it will be very interesting for many readers.

Author Response

Author’s Responses to the Comments of Reviewer 2

Comments and Suggestions for Authors:

The real significance of good analysed dates in this article is based on fundamental aspects of pathophysiological mechanisms of gustatory dysfunctions after covid and real practical recomendations which we can have after reading.  The work is well organized and logic.  I think it will be very interesting for many readers.

The author appreciates reviewer’s suggestions.
